# Radioimmune Imaging of α_4_β_7_ Integrin and TNFα for Diagnostic and Therapeutic Applications in Inflammatory Bowel Disease

**DOI:** 10.3390/pharmaceutics15030817

**Published:** 2023-03-02

**Authors:** Alberto Signore, Rita Bonfiglio, Michela Varani, Filippo Galli, Giuseppe Campagna, Manuel Desco, Lorena Cussó, Maurizio Mattei, Andreas Wunder, Filippo Borri, Maria T. Lupo, Elena Bonanno

**Affiliations:** 1Nuclear Medicine Unit, Department of Medical-Surgical Sciences and of Translational Medicine, Faculty of Medicine and Psychology, “Sapienza” University of Rome, 00185 Rome, Italy; 2Department of Experimental Medicine, University of Rome “Tor Vergata”, Via Montpellier 1, 00133 Rome, Italy; 3Unidad de Medicina y Cirugía Experimenta, Instituto de Investigación Sanitaria Gregorio Marañón, 28007 Madrid, Spain; 4Unidad de Imagen Avanzada, Centro Nacional de Investigaciones Cardiovasculares (CNIC), 28029 Madrid, Spain; 5CIBER de Salud Mental, Instituto de Salud Carlos III, 28029 Madrid, Spain; 6Departamento de Bioingeniería, Universidad Carlos III de Madrid, 28903 Madrid, Spain; 7Interdepartmental Service Centre—Station for Animal Technology, University of Rome “Tor Vergata”, 00133 Rome, Italy; 8Department of Biology, University of Rome “Tor Vergata”, 00133 Rome, Italy; 9Translational Medicine and Clinical Pharmacology, Boehringer Ingelheim Pharma GmbH & Co. KG, 88400 Biberach an der Riß, Germany; 10Unit of Pathological Anatomy, Department of Oncology, USL Toscana Sud-Est, San Donato Hospital, 52100 Arezzo, Italy; 11Unit of Pharmacy, University Hospital Sant’Andrea, 00185 Rome, Italy

**Keywords:** inflammatory bowel disease, DSS-mouse model, TNFα, α_4_β_7_ integrin, imaging inflammation

## Abstract

Imaging using radiolabelled monoclonal antibodies can provide, non-invasively, molecular information which allows for the planning of the best treatment and for monitoring the therapeutic response in cancer, as well as in chronic inflammatory diseases. In the present study, our main goal was to evaluate if a pre-therapy scan with radiolabelled anti-α_4_β_7_ integrin or radiolabelled anti-TNFα mAb could predict therapeutic outcome with unlabelled anti-α_4_β_7_ integrin or anti-TNFα mAb. To this aim, we developed two radiopharmaceuticals to study the expression of therapeutic targets for inflammatory bowel diseases (IBD), to be used for therapy decision making. Both anti-α_4_β_7_ integrin and anti-TNFα mAbs were successfully radiolabelled with technetium-99m with high labelling efficiency and stability. Dextran sulfate sodium (DSS)-induced colitis was used as a model for murine IBD and the bowel uptake of radiolabelled mAbs was evaluated ex vivo and in vivo by planar and SPECT/CT images. These studies allowed us to define best imaging strategy and to validate the specificity of mAb binding in vivo to their targets. Bowel uptake in four different regions was compared to immunohistochemistry (IHC) score (partial and global). Then, to evaluate the biomarker expression prior to therapy administration, in initial IBD, another group of DSS-treated mice was injected with radiolabelled mAb on day 2 of DSS administration (to quantify the presence of the target in the bowel) and then injected with a single therapeutic dose of unlabelled anti-α_4_β_7_ integrin or anti-TNFα mAb. Good correlation was demonstrated between bowel uptake of radiolabelled mAb and immunohistochemistry (IHC) score, both in vivo and ex vivo. Mice treated with unlabelled α_4_β_7_ integrin and anti-TNFα showed an inverse correlation between the bowel uptake of radiolabelled mAb and the histological score after therapy, proving that only mice with high α_4_β_7_ integrin or TNFα expression will benefit of therapy with unlabelled mAb.

## 1. Introduction

Crohn’s disease (CD) and ulcerative colitis (UC) are chronic inflammatory conditions of the gastrointestinal tract known as inflammatory bowel diseases (IBD). Their aetiology is still unclear, but the involvement of genetics, intestinal microbiota and immune system response drive the pathogenic mechanism [1].

Management of IBD patients is often difficult due to poor correlation between clinical symptoms and diagnostic techniques such as imaging or histology [2]. Endoscopy and colonoscopy are the gold standard for early diagnosis, but they are invasive techniques, especially for long-term treatment follow-up [3]. 

Hence, non-invasive imaging techniques would be preferred for patient management and therapy decision-making. In particular, diagnostic approaches like magnetic resonance imaging (MRI), computed tomography (CT), ultrasound (US) and nuclear medicine imaging, such as positron emission tomography (PET) and single photon emission tomography (SPECT), allow a non-invasive evaluation of bowel inflammation [4,5,6]. IBD is characterized by mucosal inflammation that leads to irreversible gastrointestinal damage, mediated by pro-inflammatory cytokines such as interleukin-12 (IL-12), tumor necrosis factor-α (TNFα), α_4_β_7_ integrin as well as immune system cells like T-lymphocytes, regulatory cells or natural killer cells (NK) [7]. These molecules are differently distributed within the intestinal wall, at different disease stages. They could represent key biomarkers to evaluate and predict the mucosal damage [8]. 

These molecules might be targeted through the use of monoclonal antibodies (mAbs) labelled with PET or SPECT isotopes. Radiolabelled mAbs can be exploited in hybrid imaging that combines radiological and nuclear medicine techniques (PET/CT, PET/MRI and SPECT/CT), providing an in vivo evaluation of intestinal damage at cellular and molecular level. Thus, molecular imaging could help to better understand the pathophysiology of the disease in relation to histological analysis to develop new targeted therapies. 

This can be achieved by using specific preclinical models that have already proven to be of great help in understanding the pathogenesis and the molecular pathways involved in the disease. In the literature, there are many reports that describe the importance of IBD animal models, including chemically induced, genetically engineered, cell-transfer models and congenic models. 

Histopathological aspects of each one and similarities with human disease have also been discussed. Among the many available models, we selected the Dextran Sulfate Sodium (DSS) model because it is particularly useful for drug screening studies and allows for the study of mucosal inflammation in different stages of the disease. Indeed, it is possible to tune DSS concentration and duration of administration to obtain a more acute or chronic conditions [9]. We have already demonstrated that this model can be helpful to evaluate different biomarkers involved in IBD with high reproducibility [10].

Our first aim was to show that TNFα and α_4_β_7_ integrin, two well-known antigens targeted for CD therapy, can be present in different amounts and different parts of the bowel in DSS mice and can be detected and quantified by nuclear medicine imaging. Furthermore, anti-TNFα and anti-α_4_β_7_ integrin mAbs were chosen as therapeutic biomarkers and for therapy decision-making in relation to the current treatments with mAbs such as Infliximab [11] or Vedolizumab [12]. As a second aim, we wanted to investigate if, in initial IBD, imaging before therapy, could predict the outcome of therapy on an individual basis. This will support the importance of a pre-therapy scan with radiolabelled antibodies in order to evaluate the appropriateness of therapy and to predict the success of therapy in relation to the expression of biomarkers in the bowel.

## 2. Materials and Methods

### 2.1. Ethical Statement

Experimental procedures were previously approved by the OPBA (the institutional animal welfare body) of the University of Rome “Tor Vergata” and were then authorized by the Ministry of Health with identification code 188/2016-PR of 22 February 2016. The study was carried out in accordance with the Italian and European regulations on the protection of animals used for scientific purposes (D.L.vo 26/2014; C.E. 63/2010). Mice were housed in the animal facility of Hospital General Universitario Gregorio Marañón, Madrid (HGUGM), Spain (ES280790000087). All animal procedures conformed to EU Directive 2010/63EU and national regulations (RD 53/2013). All animal procedures were approved by the HGUGM Animal Experimentation Ethics Committee the local Ethics Committees and by the Animal Protection Board of the Comunidad Autónoma de Madrid.

### 2.2. DSS Mouse Model

C57BL/6 female mice (8 to 10 weeks of age) were purchased from Harlan Laboratories, srl (Lesmo, MB, Italy). Animals were housed in the animal care facility of the University of Rome “Tor Vergata” in collective cages at 20 ± 2 °C under a 12 h light/dark cycle, with 4RF25 GLP food (Mucedola s.r.l., Settimo Milanese, MI, Italy) and water was provided ad libitum. They were allowed to acclimate to these conditions for at least 7 days before performing any experiments. Colitis was induced, as previously described [10], in C57BL/6 female mice by administration in drinking water of 2% DSS for 5 days, followed by 2 days drinking water without DSS. Control mice received normal drinking water throughout. Body weight was monitored during the whole study. 

### 2.3. Histopathological Studies

Mice were sacrificed by cervical dislocation, colon was excisedexamined for length (from anus to caecum) and immediately fixed in a 10% (*w*/*v*) formalin solution. After 24 h, fixed colons were divided into four equal parts and feces were removed from each portion as previously described [10]. Briefly, the colon was divided into caecum-ascending colon (A), transverse colon (T), descending colon (D) and sigmoid-rectum (R). Each part was then divided into 4 smaller parts of 5 mm and paraffin was embedded. Each part was cut into smaller rings and analysed histologically (several sections 4 µm thick). Slices were stained by hematoxylin and eosin (H&E) [13] and scored, as previously described, from 0 to 3 [10]. Immunohistochemical analyses (IHC) was performed to assess the α_4_β_7_ integrin and TNF-α distribution in each section [10]. Briefly, antigen retrieval was performed by the pressure-cooker method (2100 Retriever; Aptum Biologics Ltd., Southampton, United Kingdom) on 3 μm-thick paraffin sections using EDTA citrate pH 7.8 (α_4_β_7_ integrin) or citrate pH 6.0 (TNF-α) for 20 min at 120 °C. Sections were then incubated for 1 h at room temperature with primary antibodies (anti TNFα: Rat monoclonal, clone XT3.11; BioXCell, Lebanon, NH, USA, diluted 1:1000; anti-α_4_β_7_: Rat monoclonal, clone DATK32; BioXCell, Lebanon, NH, USA, diluted 1:700). Washing steps were performed with PBS/Tween20 pH 7.6. Reactions were revealed by an HRP—DAB Detection Kit (UCS Diagnostic, Rome, RM, Italy) and counterstained with hematoxylin. To analyse IHC, stained slides were digitalized (Iscan Coreo, Ventana, Tucson, AZ, USA) and images were captured by using ImageView Software. IHC positivity was scored from 0 to 3 [10]. Considering the intensity and the number of positive cells detected following the immunoreactions for TNFα, the score (0–3) was attributed by establishing different cut-offs (score 0: positive cells ≤ 2; score 1: 3 < positive cells ≤ 50; score 2 (51 < positive cells ≤ 200), score 3 (positive cells > 200). To assess the background of immuno-staining, for each reaction, we included a negative control by incubating the sections with secondary antibodies (HRP) and a detection system (DAB). According to the manufacturer’s instructions, reactions were set up by using specific control tissues.

### 2.4. mAbs and Cell Lines

Anti-α_4_β_7_ integrin (clone: DATK32), anti-TNFα (clone: XT3.11) and IgG2a control isotype (clone: 2A3) mAbs were purchased from BioXCell (West Lebanon, NH, USA). J774A.1 monocyte-macrophage cell line (ATCC^®^ TIB67™) producing TNFα and TK1 T-lymphocyte cell line expressing α_4_β_7_ integrin (ATCC^®^ CRL-2396™) were purchased from ATCC (Manassas, VA, USA).

### 2.5. Radiolabelling of Anti-α_4_β_7_ Integrin and Anti-TNFα mAbs

Each mAb was radiolabelled by an indirect method using succinimidyl-6-hydrazinonicotinate hydrochloride (HYNIC) (ABX advanced biochemical compounds, Radeberg, Germany). This heterobifunctional crosslinker reacts with free ε-amino groups of lysine in proteins and chelates technetium-99 m [14]. Briefly, S-HYNIC was dissolved in dimethylformamide (70 µM) and added to a vial containing the mAb (2 mg). The mixture was incubated for 1 h in the dark, at room temperature. After incubation, the solutions were purified by HiTrap Gel Filtration columns (GE Healthcare, Waukesha, WI, USA) using Phosphate Buffer Saline (PBS) as eluent. The amount of purified conjugated mAbs was determined by Bicinchoninic acid (BCA) protein assay (Thermo Scientific, Waltham, MA, USA) in relation to a standard curve. Different HYNIC:protein molar ratios (5:1, 10:1 and 20:1) were tested to optimize the conjugation procedure. Then, 100 µg of each conjugated mAb was labelled with 185–370 MBq of freshly eluted ^99m^TcO_4_^−^ (100 µL NaCl 0.9%). Variable amount of co-ligand tricine (0.5–12.5 mg) and reducing agent stannous chloride (SnCl_2_) in a range between 10 µg and 1 mg were used to obtain the best labelling conditions. Therefore, tricine (Sigma-Aldrich, St. Louis, MO, USA) was dissolved in distilled water and SnCl_2_ (Sigma-Aldrich, St. Louis, MO, USA) in purged HCl 0.1 M (10 mg/mL). The reaction solution was incubated for 15 min at room temperature and the labelling efficiency (LE) and colloids percentage were evaluated by quality controls. 

### 2.6. Quality Controls and In Vitro Binding Studies

LE and colloids percentage were evaluated by instant thin layer chromatography (ITLC) and HPLC. ITLC analyses were performed using silica gel strips (Pall LifeSciences, Port Washington, NY, USA) as stationary phase, and NaCl 0.9% or NH_3_:EtOH:H_2_O (1:3:5) solution as mobile phase to determine free pertechnetate (Rf = 0.9) and colloids (Rf = 0.1), respectively. The strips were analysed by a radio-scanner (Bioscan, Inc., Poway, CA, USA) and each species was determined. HPLC was performed with a Gilson system, using a size exclusion chromatography column (Yarra 3 µm SEC-2000 column, Phenomenex, Torrance, CA, USA) and 100 mM sodium phosphate in water pH 6.8 + 0.025% NaN as mobile phase.

After radiolabelling, the stability and integrity of radiolabelled mAbs were evaluated performing stability assay, cysteine challenge and Sodium-dodecyl-sulphate polyacrylamide gel electrophoresis (SDS-PAGE), respectively.

Stability assay was performed adding 100 µL of radiolabelled mAb to 900 µL of freshly human blood serum or NaCl 0.9%. Solutions were incubated at 37 °C and the radiochemical purity was evaluated at 1, 3, 6 and 24 h.

Cysteine challenge was performed adding 100 µL of radiolabelled mAb to 900 µL of serial dilutions of cysteine. The test tubes were incubated at 37 °C for 1 h and the radiochemical purity was evaluated by ITLC. In vitro binding assays were performed to evaluate binding kinetics and affinity of mAbs for their receptor. For this purpose, saturation and kinetic binding studies were performed on cancer human cell lines expressing membrane bound α_4_β_7_ integrin or TNFα.

^99m^Tc-anti-TNFα kinetic binding studies were performed using J774A.1 monocyte-macrophage cell line (ATCC^®^ TIB67™), cultured using Dulbecco’s Modified Eagle’s Medium (DMEM; EuroClone, Italy) with 10% Fetal Bovine Serum (FBS; Gibco, Waltham, MA, USA) and Penicillin-Streptomycin-Glutamine (PSG; Gibco, Waltham, MA, USA) to a final concentration of 1%. In this assay, 10^6^ cells (10 mL) were seeded in the lower part of a tilted petri dish and placed in a humified incubator to let them firmly adhere (37 °C, 5% CO_2_). After 24 h, 10 µg of lipopolysaccharide (LPS; Sigma-Aldrich) were added and the dish was incubated again for 2 h to stimulate TNFα production. After incubation, the medium was removed and ^99m^Tc-anti-TNFα mAb (15.8 nM) was added to petri dish to start LigandTracer^®^ measurement. Binding kinetic association was evaluated for 75 min, whereas dissociation rate was evaluated for 60 min.

Since TK-1 cells do not grow in adhesion, a traditional saturation binding assay was performed for the ^99m^Tc-anti-α_4_β_7_ mAb, as described elsewhere [15]. 

### 2.7. Biodistribution Studies

The biodistribution of radiolabelled mAb was determined in healthy mice (*n* = 6 per radiopharmaceutical) and in DSS-mice (*n* = 6 per radiopharmaceutical). Each group received an intravenous injection in the lateral tail vein of radiolabelled anti-α_4_β_7_ integrin, or anti-TNFα or control isotype (3.7 MBq in 100 µL NaCl 0.9%), respectively. Images were acquired under anaesthesia using a high resolution γ-camera (Li-Tech) at 6 and 24 h. After each time point, three animals were euthanized to collect blood and major organs for ex vivo studies. Each organ was weighted and radioactivity was determined with a single well gamma-counter (PerkinElmer, Waltham, MA, USA). Colons were processed as described above. Each colon ring, was weighted and counted for radioactivity with a single-well gamma counter. After ex vivo studies, histopathological data from HE and IHC for α_4_β_7_ integrin and TNFα were finally correlated with radioactivity findings. On planar images, target-to-background (T/B) ratios were calculated by drawing a ROI over the large bowel (target) and muscle (background).

### 2.8. Targeting Studies in DSS Mice

In vivo studies in DSS mice were performed to evaluate the expression of therapeutic targets by imaging and by ex vivo counting. As control radiopharmaceutical we used a ^99m^Tc-labelled non-specific isotype mAb (IgG2). Data were also correlated with HE and IHC scores. Mice were divided into three groups, one per each mAb. Each group received an intravenous injection in the lateral tail vein of either radiolabelled anti-α_4_β_7_ integrin, anti-TNFα or control isotype mAb (3.7 MBq) (*n* = 10 per group). Images were acquired under anaesthesia with high resolution γ-camera (Li-Tech) at 6 and 24 h. After each time point, five animals were euthanized to collect blood and major organs for ex vivo studies. Each organ was weighted and radioactivity was determined with a single well gamma-counter (PerkinElmer, Waltham, MA, USA). Colons were then formalin-fixed and processed as described above. In two out of five mice, ex vivo imaging of each segment was performed at 6 and 24 h. Each segment was cut again into smaller rings (with an average of 5 rings per tract) that were finally weighted and counted with in the gamma counter to evaluate the uptake of injected radiopharmaceutical in the different portions of the large bowel. After ex vivo studies, histology was performed on each ring to correlate HE and IHC scores with radioactivity findings. On planar images, T/B ratios were calculated by drawing a ROI over the large bowel (target) and muscle (background) to compare to T/B ratios of DSS-non-treated mice. SPECT/CT imaging was performed in additional three groups (*n* = 4 per group) and imaged at 24 h p.i. using a small-animal SPECT scanner (μSPECT, MILABS, Houten, the Netherlands) and a preclinical CT system (Super Argus, SEDECAL, Madrid, Spain). at 6 and 24 h. To co-register the SPECT and CT images, each animal was placed on an in-house multimodal bed surrounded by three noncoplanar capillaries filled with a mixture of ^99m^Tc and Iopamiro (Bracco Imaging S.p.A, Milan, Italy).

### 2.9. Therapy Outcome Prediction through Nuclear Medicine Imaging

After proving the specificity of our new radiopharmaceuticals, we finally investigated the possibility to use ^99m^Tc-anti-TNFα or ^99m^Tc-anti-α_4_β_7_ for predicting the efficacy of specific therapy with unlabelled mAbs, in mice at beginning of IBD induction. Twenty C57BL/6 mice were divided into two groups (n = 10 per group) and colitis was induced as described above. On day 2 from the start of DSS administration, mice received an intravenous injection in the lateral tail vein of radiolabelled anti-α_4_β_7_ integrin or anti-TNFα and planar gamma-camera images were acquired under anaesthesia with a high resolution γ-camera for small animal imaging. After image acquisition at 6 h p.i., 300 µg of anti-α_4_β_7_ integrin or anti-TNFα mAb were injected intraperitoneally. Body weight was monitored during the whole study. From day 3 to 5, mice continued to be treated with DSS. On day 7, mice were euthanized to collect colons that were processed as described above, followed by HE. On planar images, T/B ratios were calculated by drawing a ROI over the large bowel (as target) and thigh muscle (as background).

### 2.10. Statistical Analysis

Continuous variables were presented as mean ± standard deviation (SD) and 95% confidence interval (95%CI). The normality of the continuous variables and of the residuals was assessed by Shapiro–Wilk test. Data analysis of the expression of α_4_β_7_ integrin and TNFα between control vs. DSS-treated mice vs. therapy was performed using a General Linear Model (GLM). Homoscedasticity was evaluated checking the plot of residuals vs. fitted plot. Post-hoc analysis was performed by Tukey method. Comparisons between control vs. DSS-treated mice of injected dose per gram (%ID/g) both of the expression of α_4_β_7_ integrin and TNFα were performed by Student’s *t* test. Pearson’s correlation was used in the presence of normality of continuous variables while Spearman’s correlation was used when normality failed. 

A *p* value < 0.05 was considered statistically detectable. All statistical analyses were performed using SAS v.9.4 and JMP PRO v. 16.0 (SAS Institute Inc., Cary, NC, USA).

## 3. Results

### 3.1. Histology

As we previously published, here, we confirm that the distal colon was mainly involved in mucosal damage in terms of architectural and epithelial damage. In the entire colon, the expression of TNFα was found higher in DSS-treated mice than controls [1.83 ± 0.32, 95%CI: 1.43 to 2.24 vs. 0.71 ± 0.42; 95%CI: (−0.32 to 1.75), *p* < 0.0001], with prevalent TNFα–positive cells in descending and sigma-rectum tracts; while α_4_β_7_ integrin expression showed no difference between the two groups [2.14 ± 1.08, 95%CI: 0.80 to 3.49 vs. 1.57 ± 0.27, 95%CI: 0.89 to 2.52; *p* = 0.41], (Figure 1).

### 3.2. Radiolabelling and QCs

The highest labelling efficiency was obtained using a HYNIC:mAb molar ratio of 5:1, 28 µmol of tricine, 0.1 µmol of SnCl_2_ and 185-370 MBq of ^99m^TcO_4_^−^. After the last purification step, radiochemical purity of all mAbs was > 96%, with a negligible amount of colloids (<5%). Biochemical quality controls showed a retained integrity of each radiolabelled mAb and all radiopharmaceuticals are stable in human serum, NaCl 0.9% and in cysteine solution up to 24 h (radiochemical purity > 95%).

### 3.3. In Vitro Binding Assays

^99m^Tc-anti-TNFα kinetic binding studies showed that mAb was able to bind to TNFα positive cells, reaching a plateau at 60 min. Retention studies showed a rapid off-rate with a dissociation constant equal to 5.69 nM (Figure 2). Saturation binding assay on TK-1 showed that also the radiolabelled anti-α_4_β_7_ integrin mAb was still able to bind to its target with a Kd of 44 nM (Figure 3).

### 3.4. Biodistribution Studies in Mice

Biodistribution studies of ^99m^Tc-anti-α_4_β_7_ integrin, ^99m^Tc-anti-TNFα and ^99m^Tc-control isotype mAb are summarized in Appendix A. The ^99m^Tc-anti-α_4_β_7_ integrin showed a prevalent spleen uptake and a mild signal from liver and kidneys at 6 and 24 h. At 24 h, high circulating activity was observed up to 6 h, with a decrease within 24 h. In DSS-treated mice, ^99m^Tc-anti-α_4_β_7_ integrin showed a higher uptake in the large bowel in comparison to control mice.

The ^99m^Tc-anti-TNFα mAb showed hepatic uptake due to metabolism that persists up to 24 h with a lower signal from the spleen. High circulating activity was observed in the first 6 h with a 50% decrease at 24 h. In DSS-treated mice, a similar ^99m^Tc-anti-TNFα biodistribution might be observed at 6 and 24 h, except for colon uptake that was higher than in control mice in terms of percentage of injected dose per gram (%ID/g) at 24 h. The ^99m^Tc-control isotype showed a main hepatic metabolism at each time point and high circulating activity in the first 6 h that persists up to 24 h. In DSS-treated mice, ^99m^Tc-control isotype mAb was prevalently excreted from liver with a slight signal from spleen and kidneys at 6 and 24 h. Moreover, high circulating activity can be observed at 6 h followed by a 50% decrease at 24 h.

### 3.5. Targeting Studies

Ex vivo counts analysis showed high and homogeneous uptake of ^99m^Tc-anti-TNFα mAb in the colon of DSS-mice with a signal reduction at 24 h (Figure 4). Statistical analysis confirmed that the difference between control and DSS colons was significant [1.69 ± 1.14; 95%CI: (0.50 to 2.89) vs. 5.05 ± 2.00, 95%CI: (3.38 to 6.73), *p* = 0.032]. 

^99m^Tc-anti-α_4_β_7_ integrin mAb showed a ubiquitous distribution between large bowel segments in DSS-mice with a prevalent uptake at level of descending colon (D) and sigma-rectum (R) segments, without statistically detectable differences versus control mice [9.36 ± 10.32, 95%CI: (−1.47 to 20.19) vs. 11.72 ± 10.04, 95%CI: (4.00 to 19.44), *p* = 0.34]. The ^99m^Tc-control mAb showed a diffuse localization in all segments of large bowel in control and DSS-treated mice without significant differences (*p* = 0.19). 

A slightly higher uptake was observed in descending colon (D) and sigma-rectum (R) segments of DSS mice (*p* = 0.56 and *p* = 0.35, respectively), likely due to non-specific leaking because of the presence of oedema. 

These findings were confirmed by ex vivo imaging of isolated organs, as shown in Figure 5.

SPECT/CT scans confirmed accumulation in the large bowel of ^99m^Tc-anti-α_4_β_7_ integrin and ^99m^Tc-anti-TNFα mAb (Figure 6).

When we correlated the radioactivity in each colon ring with the IHC score, we found a strong correlation in DSS-mice injected with ^99m^Tc-anti-α_4_β_7_ or with ^99m^Tc-anti-TNFα (r = 0.84 (0.6 to 0.95), *p* < 0.0001 and r = 0.81 (0.41 to 0.95), *p* = 0.0001, respectively), but no correlation was found when comparing radioactivity in rings of DSS-mice injected with ^99m^Tc-control isotype and IHC score of either anti-α_4_β_7_ or anti-TNFα mAbs (both *p* = ns), (Appendix A).

### 3.6. Therapeutic Studies in DSS Mice

DSS treatment induced a significant increase in the IHC score of both biomarkers, as compared to control groups (Table 1). 

On the other hand, the therapy groups treated with unlabelled mAbs showed a significant drop of the expression of α_4_β_7_ integrin and TNFα, as revealed by IHC studies (Table 1). 

T/B ratios of the large bowel, calculated on planar scintigraphic images, before therapy, inversely correlated with the histological score at the end of therapy, meaning that the highest was the presence of α_4_β_7_ integrin or TNFα before therapy, the most successful was the therapy resulting in a lower histological score (Figure 7).

## 4. Discussion

The etiology of IBDs is still unclear, but involvement of immune cells and soluble factors has been demonstrated. The intestinal inflammatory condition determines an over-expression of inflammatory factors that might be used as therapeutic biomarkers. Nowadays, many biological therapies are available against specific targets [16,17], but not all patients successfully respond to these treatments. For these reasons, the evaluation of biomarker expression prior to therapy administration, through non-invasive techniques, is essential for patient management, therapy decision-making and treatment follow-up. Given the complexity of IBD pathophysiology, it is difficult to develop an animal model that fully reflects human disease. Among the many models described in the literature, the DSS mouse is the most comparable for human ulcerative colitis [18]. In a previous paper, we already described an extensive histopathological characterization of the DSS mouse model, focusing attention on the relevant biomarker and targets involved in IBD for drug discovery [10]. We found that α_4_β_7_ integrin and TNFα are highly expressed in this model since the beginning of DSS treatment, despite some differences in the extent of expression among mice were observed. After 5 days of DSS administration, all mice developed severe colitis that could be prevented by treating them with unlabelled anti-α_4_β_7_ integrin or anti-TNFα mAb at therapeutic doses.

Therefore, in the present study, our main goal was to evaluate if a pre-therapy scan with radiolabelled anti-α_4_β_7_ integrin or radiolabelled anti-TNFα mAb could predict therapeutic outcome with unlabelled anti-α_4_β_7_ integrin or anti-TNFα mAb. 

To this aim, our experiments were designed to investigate if mice with early expression of target molecules (as detected by scan at day 2 of DSS) were more or less responsive to 5 days’ treatment (as detected by histology at day 7). 

We selected two human therapeutic biomarkers (α_4_β_7_ integrin and TNFα) and developed two radiopharmaceuticals to study their expression in DSS-treated mice. As anti-α_4_β_7_ integrin or anti-TNFα mAb, we chose mouse-specific antibodies to avoid poor affinity of anti-human antibodies, as previously described [19], despite other authors having successfully used radiolabelled infliximab in mice [20]. Control isotype mAb was selected as control and radiolabelled with ^99m^Tc. Each mAb was efficiently radiolabelled with high labelling efficiency (>95%) and high specific activity (2.9 × 10^5^ MBq/µmol), as shown by quality controls (Figure 3). All radiopharmaceuticals showed high stability in human serum and NaCl 0.9% within 24 h, and the radiolabelling procedure did not affect the mAb structure or the binding properties, as confirmed by SDS-PAGE analysis and in vitro binding assays, respectively. The biodistribution of radiolabelled mAbs were similar to those of other radiolabelled mAbs, with typical high circulating half-life, main hepatic clearance and slight spleen uptake due to the presence of activated lymphocytes, as already reported in the literature [11]. To evaluate the clinical specificity of radiopharmaceuticals, for each group of study (five DSS vs. three control mice) each colon of each animal was cut in four equal segments (ideally corresponding to ascendant, transverse, descendant colon and sigma-rectum tracts) and, secondly, each segment was cut in several rings (of 5 mm each), on which, ex vivo radioactivity counting and IHC analysis were performed. Results showed that α_4_β_7_ integrin and TNFα markers expression increased in large bowels of DSS mice, respectively, 2.5 and 3-fold higher than in control mice. These biomarkers, detected by radiolabelled mAbs, are differentially expressed along the large bowel, suggesting a specific role of each one in IBD evaluation/management.

In particular, α_4_β_7_ integrin was rather ubiquitously concentrated in the entire large bowel, albeit predominantly towards transvers, descendant colon and sigma-rectum, as demonstrated by ex vivo studies of radiolabelled rings of each segment compared to their immunohistochemical data. Ex vivo imaging confirmed the same observations of IHC and ex vivo counts, as well as the SPECT-CT imaging. 

TNFα was mainly expressed at the level of descendant colon and sigma-rectum as reported in IHC, ex vivo and in vivo studies. IHC analysis and radioactivity findings were not always fully correlated because TNFα, being a soluble factor, could act at a distance and not be fully found at the local level.

The ^99m^Tc-control isotype mAb showed a ubiquitous localization in all segments of large bowel of both DSS and control mice, but slightly higher in DSS mice. This higher uptake in DSS-treated mice could be due to increased oedema in inflamed areas, not due to the presence of specific targets. Probably, also the higher uptake of ^99m^Tc-anti-α_4_β_7_ integrin and ^99m^Tc-anti-TNFα mAbs might be due to increased oedema in intestinal inflamed lesions, but the over-expression of specific targets may be considered the main mechanism since the uptake in vivo correlates with target expression by IHC. 

Based on preliminary experiments, for both radiopharmaceuticals, the best acquisition time was at 24 h post i.v. injection. SPECT/CT imaging supported findings obtained by planar images, although in some mice rectum-sigmoidal activation was better detected by SPECT than by planar images, due to bladder activity.

Indeed, a small degree of inter-individual variability was observed in mice despite the high reproducibility of DSS-induced IBD. This aspect makes it much more complicated to assess the state and progression of disease, to decide the best therapeutic approach and to follow its efficacy, as well as to establish standard criteria for IBD diagnosis and treatment. Imaging modalities could represent a valid and non-invasive alternative to more invasive immuno-histological evaluation or colonoscopy, making patients more comfortable. In our study, mice treated with unlabelled mAbs showed a significant decrease of the IHC scores of both TNFα and α_4_β_7_ integrin and these values inversely correlated with the uptake of radiolabelled mAbs in mice colons. Therefore, mice with high radiopharmaceutical uptake are more prone to benefit from therapy with cold TNFα or anti-α_4_β_7_ integrin mAbs. The translation of these techniques to predict or follow-up therapeutic efficacy would be of great importance.

Furthermore, another aim of this study was to validate this nuclear medicine imaging modality as an option for IBD diagnosis, alone or in combination with colonoscopy and histology. Indeed, histology and nuclear molecular are extremely different techniques and it is very difficult to correlate results of the two. Nevertheless, we found a significant correlation between images, ex vivo uptake and histology.

From the present study, it emerges that α_4_β_7_ integrin and TNFα are predominantly distributed to the last segments of the large bowel in the acute phase of disease. Although in all our ex vivo experiments, the bowel was carefully cleaned up from feces, we cannot exclude some interference of food during the in vivo studies. Understanding the position and the stage of biomarker up-regulation would be clinically relevant for therapy decision-making and the personalized treatment of patients.

Indeed, several studies investigated the efficacy of treatment with anti-TNFα and Vedolizumab in IBD patients, concluding the coexistence of non-responder patients to one of the two therapies or to both therapies [21,22,23]. Previously, anti-TNFα mAb had been investigated for scintigraphic imaging and biological therapy of CD patients, suggesting the necessity of further studies to clarify its mechanism of action [10]. Other authors have demonstrated that α_4_β_7_ integrin expression on peripheral blood lymphocytes correlates with the success of vedolizumab treatment [24]; however, the detection of this biomarker does not inform about the severity and the extent of α_4_β_7_ integrin expression in the affected bowel. Hence, the development of new radiopharmaceuticals targeting specific biomarkers is greatly needed for patient management prior to therapy and post-therapy. In line with this strategy, others have recommended fluorescent anti-α_4_β_7_ integrin to be used during colonoscopy [25] or ^64^Cu-labelled anti-α_4_β_7_ integrin for PET imaging of murine colitis [26,27], but no reports are present on 99mTc-labelled anti-α_4_β_7_ integrin mAb. On the contrary, there are several reports describing the use of ^99m^Tc-anti-TNFa mAbs in several diseases such as rheumatoid arthritis [28] and Graves’ ophthalmopathy [29].

Other studies have previously radiolabelled mAbs and peptides for the prediction of therapy response in IBD. In particular, anti-β_7_ integrin radiolabelled with ^64^Cu has been used to assess the intestinal inflammation in animal models of experimental colitis and to evaluate the disease extent and activity [27].

Therefore, this study highlights how α_4_β_7_ integrin and TNFα increased their expression in inflamed large bowels compared to control colons, but with very high variability, depending on inflammation site, disease phase and the patient-specific response to disease. 

In conclusion, the availability of tools for non-invasive IBD evaluation could help clinicians choose the best therapy for each patient, reducing treatment costs and other complications to patients.

## Figures and Tables

**Figure 1 pharmaceutics-15-00817-f001:**
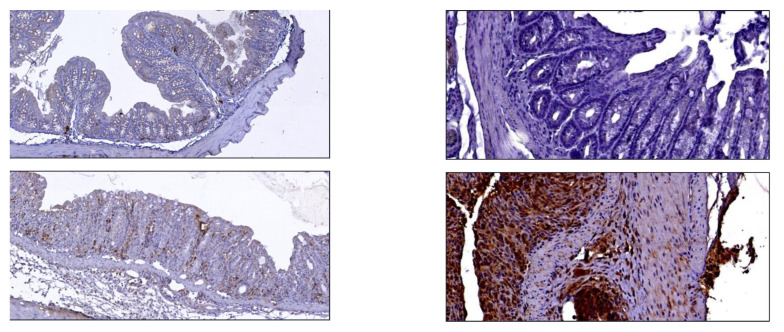
Anti-α_4_β_7_ integrin (**left**, 10×) and anti-TNFα (**right**, 20×) IHC staining of colon in control mice (**top**) vs. DSS mice (**bottom**).

**Figure 2 pharmaceutics-15-00817-f002:**
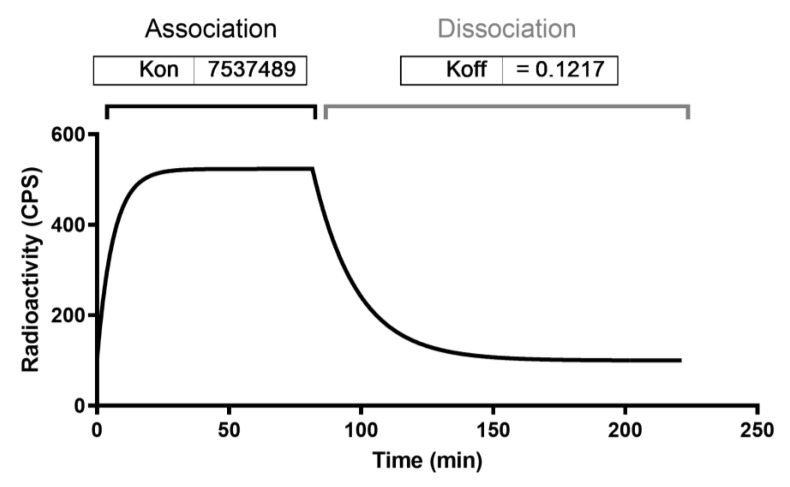
Kinetic binding assay of radiolabelled anti-TNFα mAb on J744A.1 cells.

**Figure 3 pharmaceutics-15-00817-f003:**
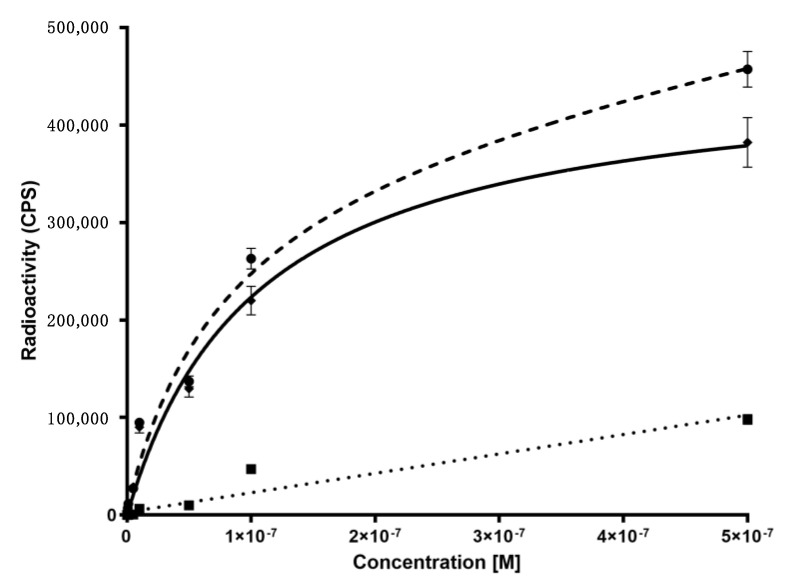
Saturation binding assay of radiolabelled anti-α_4_β_7_ integrin mAb showing total binding (dashed line), specific binding (continuous line) and non-specific binding obtained by blocking with 100-fold molar excess unlabelled anti-α_4_β_7_ integrin mAb (dotted line) on TK-1 cells.

**Figure 4 pharmaceutics-15-00817-f004:**
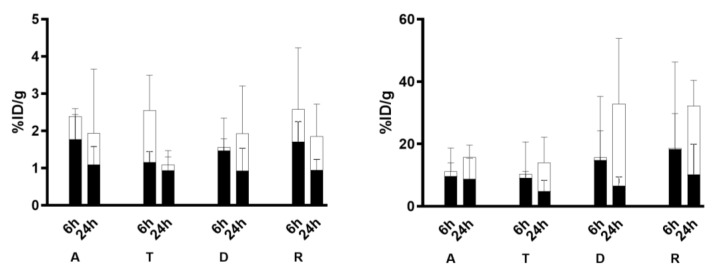
Ex vivo counts of A-T-D-R colon segments of control (black) and DSS (white) mice injected with ^99m^Tc-anti-α_4_β_7_ integrin (**left**) or ^99m^Tc-anti-TNFα (**right**). Data are expressed as %ID/g ± SD. ^99m^Tc-anti-α_4_β_7_ integrin accumulates particularly in the transvers and descending colon. ^99m^Tc-anti-TNFα accumulates particularly in the descending colon and rectum at 24 h p.i. A = Ascending colon; T = Transvers colon; D = Descending colon; R = Rectum-sigma.

**Figure 5 pharmaceutics-15-00817-f005:**
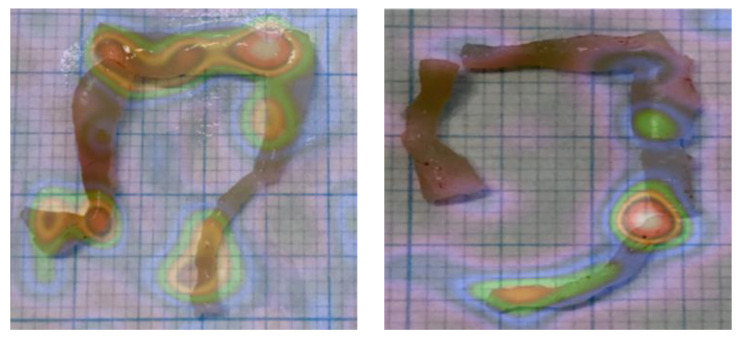
Representative ex vivo images of colons from DSS mice after injection of ^99m^Tc-anti-α_4_β_7_ (**left**) and ^99m^Tc-anti-TNFα (**right**). Similarly to Figure 4, ^99m^Tc-anti-α_4_β_7_ integrin accumulates particularly in the transvers and descending colon, while ^99m^Tc-anti-TNFα accumulates particularly in the descending colon and rectum.

**Figure 6 pharmaceutics-15-00817-f006:**
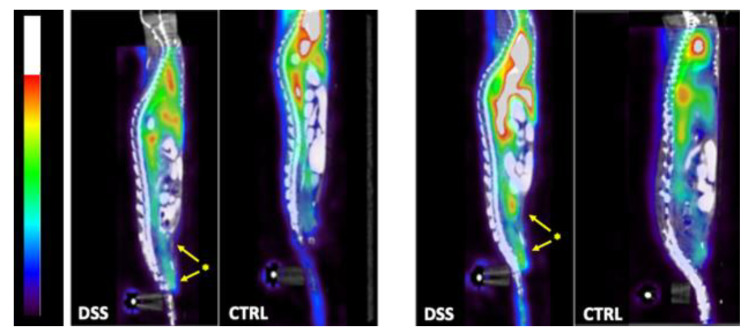
Representative sagittal SPECT/CT images of DSS and CTRL mice injected with ^99m^Tc-anti-α_4_β_7_ (**left**) or ^99m^Tc-anti-TNFα (**right**). * = large bowel activity.

**Figure 7 pharmaceutics-15-00817-f007:**
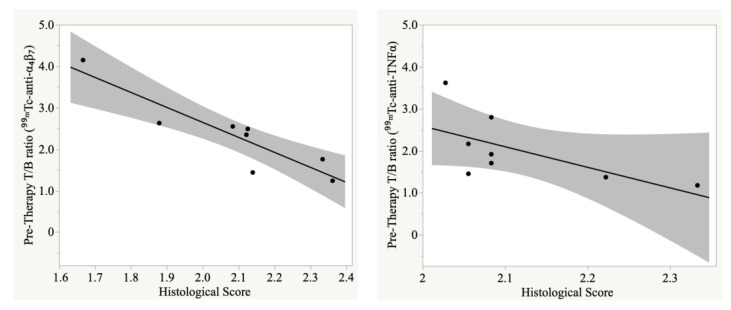
Correlation between pre-therapy T/B ratio and post-therapy histological (HE) score calculated on DSS mice injected with ^99m^Tc-anti-α_4_β_7_ (**left**) or ^99m^Tc-anti-TNFα (**right**), (r = −0.72 (−0.95 to −0.04); *p* = 0.04 and r = −0.90 (−0.98 to −0.55); *p* = 0.002, respectively). The highest is the presence of target antigen before therapy, the less is the histological damage at the end of therapy. This is more evident for mice treated with anti-α_4_β_7_ mAb. Gray area = confidence interval.

**Table 1 pharmaceutics-15-00817-t001:** IHC scores for radiolabelled mAbs in control, DSS and therapy groups.

**Anti-α_4_β_7_ Integrin**				
**Parameter**	**Control**	**DSS**	**Therapy**	
	mean ± SD; (95%CI)	mean ± SD; (95%CI)	mean ± SD; (95%CI)	
IHC	1.57 ± 0.27; (0.89 to 2.52)	2.14 ± 1.08; (0.80 to 3.49)	0.58 ± 0.25; (0.40 to 0.75)	*p* = 0.0001
Post-hoc analysis:				
IHC—(Control vs. Therapy), *p* = 0.004; (DSS vs. Therapy), *p* = 0.0002;				
**Anti-TNFα**				
**Parameter**	**Control**	**DSS**	**Therapy**	
	mean ± SD; (95%CI)	mean ± SD; (95%CI)	mean ± SD; (95%CI)	
IHC	0.71 ± 0.42; (−0.32 to 1.75)	1.83 ± 0.32; (1.43 to 2.24)	0.22 ± 0.17; (0.10 to 0.34)	*p* < 0.0001
Post-hoc analysis:				
IHC—(Control vs. DSS), *p* < 0.0001; (Control vs. Therapy), *p* = 0.029; (DSS vs. Therapy), *p* < 0.0001				

## Data Availability

All data are available upon request to alberto.signore@uniroma1.it.

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
