# Peer review of "Radioimmune Imaging of α4β7 Integrin and TNFα for Diagnostic and Therapeutic Applications in Inflammatory Bowel Disease"

_pharmaceutics, 2023, doi:10.3390/pharmaceutics15030817_

Round 1

Reviewer 1 Report

In the paper “Imaging alfa4beta7 Integrin and TNFalfa for Diagnostic and Therapeutic Purposes”, Dr Signore et al. use radiolabelled mAbs to diagnose inflammatory bowel disease (IBD) in a mouse model, comparing the results from mice with induced IBD and healthy control mice. The uptake is investigated both with immunohistochemistry (IHC) of ex vivo tissues, and with SPECT/CT of whole mice. In a second group of mice with IBD treated with therapeutic amounts of the mAbs, an inverse correlation is found between the (imaged?) uptake of mAb before treatment and the histological score after treatment, indicating that such imaging may be a predictor of which patients would benefit from treatment with the corresponding mAb, and which patients would not benefit.

Major issue:

1. For these tracers to be useful in imaging of individual patients (or individual mice), not just groups, the clinician must be able to diagnose from an individual image. This aspect should be further described. For instance, what should the reader look for in Figure 5 to distinguish uptake representing IBD from non-specific uptake of the isotype tracer? Likewise, given only one of the images for T-99m-anti-TNFalfa, what would be used to characterise the mouse as either DSS or CTRL?

Minor issues:

2. Lines 42-43: No keywords are given

3. Headline of section 2.6: Please consider adding the word “binding” to the headline (“... in vitro binding studies”) to help the reader more easily connect the methods here with the results in section 3.3.

4. Line 238: SPECT/CT imaging is introduced is line 238, but no details on how it was performed is given, e.g. which instrument was used (the Li-Tech used for planar imaging in lines 211-212)? Please give some more details.

5. Figure 4: The A-T-D-R designation is not explained in the figure legend, and does not seem to be explained in the main text, either, only under the table of the supplementary data. Please include this description in the figure legend or in the main text of the paper.

6. Figure 6: The images are presented in the order DSS and CTRL, while the figure legend uses the opposite wording: “CTRL and DSS”. To help the reader, please give wording in the same order in the figure and the figure legend.

Typographical:

Line 34: A “)” appears to be superfluous.

Lines 89-90: Please check to commas.

Lines 118-121: Please check punctuation and lineshift.

Section 2.3: In this section, alfa4beta7 is written with number in normal size, whereas elsewhere in the paper, the numbers are in superscript. Please also check line 453

Line 138: “... in 4 fields at 20x (score: ...” It is suggested to add the word “magnification”: “... at 20x magnification ...”

Line 289: “... cells reaching after 60 minutes.” Is a word missing here after “reaching”?

Lines 501-514: The figure legends are repeated at the end of the paper (with some typographical issues for the Greek letters). This appears to be superfluous, as the legend were already given along with the figures.

Author Response

Reply attached

Reviewer 2 Report

The authors evaluated two Tc-99m labelled mAbs in a murine model of IBD prior to therapy with the same antibodies. Imaging was able to detect inflammation in segments of colon. It is an interesting concept that imaging could be used not only to select patients who might benefit from mAb therapy but also predict the efficacy of that therapy.

MINOR

Page 1, Abstract, line 31: DSS should be spelled out first time in abstract

TYPOS ETC

Page 2, section 2.3, lines 116-118. Two different spellings of cecum/caecum

Page 2, section 2.3. Some funny formatting has crept in

Page 13, reference 9. “(99m)“ should be superscript 99m

Page 14, reference 22. Superscript 99m

The references are not formatted consistently, e.g. capitalization

Author Response

Reply attached

Reviewer 3 Report

The article focuses on 99mTc-labeled monoclonal antibodies with a potency to visualize inflammatory bowel disease. The authors aimed at method of radiolabeling, at evaluation of radiopharmaceutical and biological quality of preparations. The key part of the manuscript presents data on in vivo distribution of the prepared biomolecules into the inflammatory sites in the large bowels and potency of the agents in therapy of colitis. The paper is ambitious in relation to potential further preclinical and clinical development, the results of the study could be potentially contributing. However, the reviewer has serious comments on the submitted work. Without appropriate modification and corrections asked below, the manuscript in this form is not sufficiently prepared for publication.

Major comments:

1. The title of the study seems to be too general. A better version could be “Radioimmune imaging a4b7 integrin and TNFalpha for diagnostic and therapeutic applications in inflammatory bowel disease”. Of course, other possibilities to modify the title could be considered by the authors.

2. It is not clear why the authors did not use for the study the known and available monoclonal antibodies targeting TNF-alpha (infliximab, adalimumab) or alpha4beta7 integrin (vedolizumab). This fact should be commented.

3. The presented results on biodistribution studies in mice (part 3.4.) involved data in DSS-treated mice (l. 309-316). However, the relevant part of Methods (part 2.7.) contains only procedure in three groups of healthy mice (l. 208-209). This is confusing for readers and must be corrected.

4. Stability assay was performed with human blood serum. But the in vivo experiments were performed in mice. Due to potential interspecies differences, the optimal analysis should also include mouse blood serum.

5. Ten of 22 cited publications in the list of literature were published by the members of the author team. Such number can be acceptable in situation when other publication related to the subject are not available. However, several similarly oriented experimental studies exist in the available literature focused on the same aim (e.g. Yan et al. Immuno-PET Imaging of TNF-α in Colitis Using 89Zr-DFO-infliximab. Mol Pharm. 2022; 19:3632-3639; Tsopelas et al. Scintigraphic imaging of experimental colitis with technetium-99m-infliximab in the rat. Hell J Nucl Med. 2006; 9:85-9). In addition, other relevant studies closely associated with the subject of the paper are not cited in the paper (e.g. Schneider et al. Expression and function of α4β7 integrin predict the success of vedolizumab treatment in inflammatory bowel disease. Transl Res. 2022: S1931-5244(22)00222-5).

6. Although blocking competitive experiments with use of high concentrations of the unlabeled ligand are usually performed in experiments evaluating binding to the target molecules in vitro, the authors did not perform them. This fact weakens the found data on binding specificity of the tested radiolabeled MAbs to the targets. An analogical study could be eventually included into in vivo experiments. The results in Figure 4 show that all tested radiolabeled antibodies including the control IgG have analogical increase in their amount in the examined colon segments of DSS mice. It may suggest that the antibody presence has non-specific character and it is not directly associated with presence of the target molecules in the damaged colon.

7. The found data on antibody biodistribution in Table S1 show relatively lower accumulation of 99mTc-antiTNFalpha antibody in the colon in comparison with the control isotypic IgG. The findings is not discussed although it means that the used 99mTc-antiTNFalpha antibody seems to be not applicable to image colitis because their accumulation is lower as in the non-specific antibody.

8. The text presented on page 9 (l. 348-352) state that there is a correlation between radioactivity in colon rings and the ICH score. However, no data on radioactivity of rings are given in tables or figures. Therefore, this statement is not possible to verify. The missing data must be added or the statement must be deleted. 

9. Because site of interest is the large bowel, an influence of food on the obtained results of in vivo experiments cannot be excluded. The potential effect is not considered in Discussion.

10. The Discussion contains minimal number of comparisons with relevant experimental studies published previously. Instead, many redundant clinical studies are cited and discussed. The reviewer recommends a complete revision of discussion section. In addition, the text of Results and Discussion should be coordinate better to be more concise.      

Minor comments:

1. Some data presented in Introduction are not related to relevant literature. For example, the third paragraph (l. 55-65) does not include any citation even there is an important statement on the target molecules. Similarly, the following paragraph (l. 66-76) does not contain any reference although the last sentence contains formulation “In the literature, there are many reports…..”. The appropriate references must be added.

2. The fourth and fifth paragraphs in Introduction (l. 66-76) are logically connected. They should be combined into one paragraph.

3. The part “2.6. Quality control and in vitro studies” should be divided because the two described activities are completely different. Eventually, the quality control should be connected to the part 2.5.

4. Several technical corrections are necessary to be performed in Abstract:

Line 33: missing “t“ in the word “he“ must be added

Line 34: redundant parenthesis must be deleted

Line 23: a space must be inserted before the word “In“.

5. Keywords (page 1) have to be inserted.

Author Response

Reply attached

Round 2

Reviewer 3 Report

Only some minor technical corrections must be performed:

1. There is a repeated error in the description of the y-axis in Figures S2-S4 ("immunohystochmistry").

2. Line 477: Better than "others " could be to use "other authors" or "other studies".

3.  Lines 486-487: The medical term "Rheumatoid Arthritis" should be written with lower case initial terms.

Author Response

We thank the reviewer and apologies for the mistakes.

A new corrected version of the manuscript has been uploaded.

AS